# Use of Urinary Cytokine and Chemokine Levels for Identifying Bladder Conditions and Predicting Treatment Outcomes in Patients with Interstitial Cystitis/Bladder Pain Syndrome

**DOI:** 10.3390/biomedicines10051149

**Published:** 2022-05-17

**Authors:** Wan-Ru Yu, Yuan-Hong Jiang, Jia-Fong Jhang, Hann-Chorng Kuo

**Affiliations:** 1Department of Nursing, Hualien Tzu Chi Hospital, Buddhist Tzu Chi Medical Foundation, Hualien 970, Taiwan; wanzu666@gmail.com; 2Department of Urology, Hualien Tzu Chi Hospital, Buddhist Tzu Chi Medical Foundation, Buddhist Tzu Chi University, Hualien 970, Taiwan; redeemer1019@yahoo.com.tw (Y.-H.J.); alur1984@hotmail.com (J.-F.J.)

**Keywords:** interstitial cystitis, painful bladder syndrome, inflammation, cytokines, chemokines, biomarkers

## Abstract

Background: Interstitial cystitis/bladder pain syndrome (IC/BPS) is a condition causing bladder inflammation. Urinary biomarkers have been assessed as suitable for the diagnosis and treatment. This study aimed at investigating the role of urinary biomarkers in identifying bladder conditions and predicting the treatment outcome of IC/BPS. Methods: A total of 309 patients with IC/BPS and 30 controls were enrolled in this study. All patients underwent a comprehensive urological workup of symptoms, pain severity, and cystoscopic hydrodistention findings including maximal bladder capacity (MBC) and glomerulation grade. Urine samples were collected to investigate the levels of urinary cytokines and chemokines. According to MBC and glomerulation grade, patients with IC/BPS were further classified into the Hunner’s IC (HIC) and non-HIC groups. The urinary biomarkers between IC/BPS and control groups and HIC and non-HIC groups were compared. Moreover, the treatment response was graded according to global response assessment (GRA) scores, and urinary biomarker levels were analyzed based on different GRAs. Results: Patients with IC/BPS had significantly high urinary monocyte chemoattractant protein-1, eotaxin, tumor necrosis factor -alpha (TNF-α), and prostaglandin E2 levels. Significantly higher levels of urinary interleukin-8, C-X-C motif chemokine ligand 10 (CXCL 10), brain-derived neurotrophic factor, eotaxin, and regulated-on-activation, normal T-cell expressed and secreted (RANTES) were noted in HIC than those with non-HIC and controls. Among all biomarkers, TNF-α had the best sensitivity, specificity, positive predictive value, and negative predictive value. There was a significant correlation between biomarker levels and GRA. Conclusions: Significantly higher urine cytokines and chemokine levels were found in patients with IC/BPS. Most urinary biomarkers were significantly associated with MBC, glomerulation grade, and treatment outcome.

## 1. Introduction

Interstitial cystitis/bladder pain syndrome (IC/BPS) is a perplexing disease of unknown etiology. The diagnosis of IC/BPS is significantly based on clinical symptoms including bladder pain and urinary frequency, which do not resolve after medical treatment for 6 weeks to 6 months [1,2]. According to a recent consensus, the condition was classified into ulcer IC (HIC) and non-ulcer IC (non-HIC), and each disease has different clinical and cystoscopic characteristics [3]. Hunner’s lesion can be observed in HIC on cystoscopy with or without anesthesia. Meanwhile, typical glomerulations can develop after cystoscopic hydrodistention under an intravesical pressure of 80 cm H_2_O in non-HIC. The histopathological findings of IC/BPS can be characterized according to lymphocyte predominant submucosal inflammation and bladder epithelial denudation [4]. Currently, endoscopic bladder biopsy is not required for diagnosing IC/BPS based on most clinical guidelines. However, it can be used for diagnostic confirmation [2,3,5,6]. HIC and non-HIC bladders have different histopathological findings that may be correlated with clinical presentation and bladder conditions [7].

The diagnosis of IC is usually made by clinical symptoms and exclusion of specific infection, cancer, autoimmune diseases, or bladder outlet obstruction according to the U.S. National Institute of Diabetes and Digestive and Kidney Diseases definition [1]. The European Association of Urology and European Society for the Study of Interstitial Cystitis have proposed that positive histopathological findings, such as inflammatory infiltrates, have been recommended to identify patients with IC/BPS, which may be helpful in supporting the clinical diagnosis and subtyping of the condition [8]. Our previous study revealed that bladder histopathological findings are closely associated with clinical IC symptoms and are different among patient-reported treatment outcomes [9]. However, in patients with non-HIC, the cystoscopic findings after hydrodistention, such as maximal bladder capacity (MBC) and glomerulation grade, varied significantly. The rate of satisfactory outcomes was better in patients with a lower glomerulation grade and a higher MBC and was significantly worse in patients with Hunner’s lesions [10].

For decades, urologists have assessed urinary and serum biomarkers that can be suitable for diagnosing IC/BPS in patients with symptoms of frequency-urgency and bladder discomfort. Proteins expressed in the bladder tissue and urine might reflect the actual condition of diseased bladders [11]. High levels of proinflammatory proteins and distinct metabolomes have been detected in urine and serum of patients with IC/BPS. This indicated that the urinary bladder is mainly involved IC/BPS [12,13]. Based on more recent studies, patients classified as the European Society for the Study of Interstitial Cystitis (ESSIC) type 2 IC/BPS had significantly higher levels of some urine cytokines. Further, a high sensitivity of IC/BPS diagnosis was noted in the patients with an elevated urinary level of normal T-cell expressed and secreted (RANTES), macrophage inflammatory protein-1 beta (MIP-1β), and interleukin (IL)-8, and they had a high specificity in urine levels of monocyte chemoattractant protein-1 (MCP-1), C-X-C motif chemokine ligand 10 (CXCL 10), and eotaxin. These urinary cytokines/chemokines were significantly correlated with clinical symptoms and cystoscopic findings [14]. However, the clinical roles of these urinary cytokines and chemokines on IC/BPS have not been fully elucidated. The current study was aimed to investigate the usefulness of these urinary cytokines and chemokines for identifying bladder conditions and predicting the treatment outcomes of IC/BPS.

## 2. Materials and Methods

A total of 309 patients including 261 women and 48 men with confirmed diagnosis of IC/BPS were consecutively enrolled in this study, from February 2010 to December 2021. The diagnostic criteria for IC/BPS were based on the ESSIC guidelines, with the exclusion of similar diseases [1,8]. This study had been approved by the Institutional Review Board and Ethics Committee of the Hospital (IRB: 105-25-B, 105-31-A, 107-175-A). All participants had been involved in different clinical trials for treatment of IC/BPS. Patients had been informed about the rationale of the study, and informed consents had been obtained. However, the need for informed consent was waived if urine samples were collected in previous clinical trials.

Patients were admitted for cystoscopic hydrodistention under general anesthesia. Hydrodistention was performed under an intravesical pressure of 80 cm H_2_O for 10 min. Next, the bladder was evacuated gradually and was cautiously inspected for the development of petechia, glomerulations, splotch hemorrhage, mucosal fissures, or Hunner’s lesion [3]. According to appearance, the grade of glomerulation was: 0: none, 1: less than half of the bladder wall, 2: more than half of the bladder wall, and 3: severe waterfall bleeding. HIC was defined as the presence of Hunner’s lesions with or without glomerulation (Figure 1). In total, 30 women with genuine stress urinary incontinence but without other storage or voiding dysfunctions were included in the control group. The detailed inclusion and exclusion criteria were similar to those in our previous study [15].

All patients with IC/BPS were assessed by the O’Leary–Saint symptom score (OSS) including IC symptom index (ICSI), the IC problem index (ICPI), and the visual analog scale (VAS) for bladder pain. The grade of glomerulation and MBC during cystoscopic hydrodistention were also recorded. Based on the glomerulation grade and MBC, non-HIC was classified into four clinical subtypes, which were as follows: (1) glomerulation grade of ≤1, MBC of ≥760 mL; (2) glomerulation grade of ≤1, MBC of <760 mL; (3) glomerulation grade of ≥2, MBC of ≥760 mL; and (4) glomerulation grade of ≥2, MBC of <760 mL [10].

### 2.1. Treatment and Outcome Assessment

After cystoscopic HD, patients with IC/BPS were consecutively treated with bladder-targeting medications for pain, such as non-steroidal anti-inflammatory drugs, agent of Cox-2 inhibitors, beta-3 adrenoceptor agonists, antimuscarinics, hyaluronic acid intravesical instillations, botulinum toxin A injections, or platelet-rich plasma injections. Moreover, they were followed-up regularly at the outpatient clinic. All patients had received at least three lines of treatment, including oral medication, intravesical instillation, and intravesical injection. Patients with Hunner’s lesion were further treated with electrocauterization, laser ablation, or partial cystectomy with or without bladder augmentation according to the outcome of the initial treatment. If the patients had a satisfactory response to any type of treatment, they would not receive the next-line therapy. The long-term treatment outcome was assessed according to the patients’ self-reported global response assessment (GRA) scores (from −3, significantly worse; to +3, significantly improved) [16]. A satisfactory treatment outcome was defined as GRA scores of +2 or +3. Meanwhile, an unsatisfactory outcome was defined as all other scores. Receiver operating characteristic (ROC) analysis was performed to find the optimal cutoff value of each urinary biomarker that could predict a successful treatment outcome (i.e., GRA score of ≥2).

### 2.2. Urinary Biomarker Investigation

The procedures of urine cytokines and chemokines investigation were in accordance with our previous study [14]. In brief, a total of 50 mL urine samples were collected before cystoscopic hydrodistention in all patients and controls. The urine samples were obtained by self-voiding when patients had a full bladder. Urine samples with confirmed urinary tract infection should be excluded. The urine samples were placed immediately on ice before transferring to laboratory. The urine samples were then centrifuged at 1800× *g* for 10 min at 4 °C. The supernatant was preserved in a freezer at −80 °C. The frozen urine samples were centrifuged at 12,000× *g* for 15 min at 4 °C before further analyses were performed, and the supernatants were used for subsequent measurements.

### 2.3. Cytokine and Chemokine Assay

In this study, we used the commercial microspheres with the Milliplex^®^ Human cytokine/chemokine magnetic bead-based panel kit (Millipore, Darmstadt, Germany) to assay the inflammation-related urinary cytokines and chemokines. According to urinary cytokines and chemokines that were previously considered as significant in the diagnosis of IC/BPS patients, we selected 10 targeted analytes, including IL-8, CXCL 10, MCP-1, brain-derived neurotrophic factor (BDNF), eotaxin, IL-6, MIP-1β, RANTES, tumor necrosis factor-alpha (TNF-α), and prostaglandin E2 (PGE2). Then, these analytes were measured by the multiplex kit (the catalog number: HCYTMAG-60K-PX30). The procedures to measure these urinary cytokines and chemokines were based on the manufacturer’s instructions and reported previously [14,16]. The median fluorescence intensity of each cytokine/chemokine target was recorded and analyzed to calculate the individual corresponding cytokine/chemokine concentration in urinary samples.

### 2.4. Statistical Analysis

Continuous variables were expressed as means ± standard deviations and categorical data as numbers and percentages. The urinary cytokine and chemokine levels between the IC/BPS and control groups, and the HIC and non-HIC groups were analyzed using analysis of variance. The cytokines with a mean value below the minimum detectable concentrations were not included in the final analysis. Receiver operating characteristic (ROC) curves and the cutoff values of each cytokine and chemokine were generated to assess the sensitivity, specificity, positive predictive value (PPV), and negative predictive value (NPV) of each analyte in distinguishing IC/BPS patients from controls, as well as identifying HIC patients from non-HIC patients. The Statistical Package for the Social Sciences software for Windows version 20.0 (IBM Corp., Armonk, NY, USA) was used for statistical analysis, and the *p* values of <0.05 were considered statistically significant.

## 3. Results

In total, 309 patients (261 women and 48 men) with IC/BPS and 30 controls (all women) were enrolled in this study. Their mean age were 53.1 ± 13.4 and 57.7 ± 10.1 years, respectively (*p* = 0.068). At enrolment, the mean ICSI was 11.1 ± 4.56; ICPI was 10.9 ± 3.82; the VAS score was 4.48 ± 2.87; and MBC was 721.8 ± 186.5 mL. After cystoscopic hydrodistention, five subgroups were classified in IC/BPS patients according to their MBC and grade of glomerulation. The groups were as follows: (1) MBC of >760 mL and glomerulation of ≤1 (*n* =85); (2) MBC of ≤760 mL and glomerulation of ≤1 (*n* = 70); (3) MBC of >760 mL and glomerulation of ≥2 (*n* = 41); (4) MBC of ≤760 mL and glomerulation of ≥2 (*n* = 89); and (5) HIC (*n* = 24).

Table 1 depicts the urinary biomarker levels of overall patients with non-HIC and HIC and the controls. Significantly higher urinary levels of IL-8, CXCL 10, BDNF, eotaxin, and RANTES were noted than the levels in non-HIC patients and controls. Significantly higher urinary levels of MCP-1, eotaxin, TNF-α, and PGE2 were noted in non-HIC patients than those in controls.

When we divided patients into male and female IC/BPS patients, and compared the urine biomarkers between male and female patients, a lower level of IL-8 and CXCL 10 and a higher level of PGE2 were found in male non-HIC patients than those in female patients. However, there was no significant difference in all urine biomarkers between male and female HIC patients. (Table 2) The differences of urinary levels of biomarkers compared with those of controls were not different between overall and male or female IC/BPS patients. Therefore, we did further analysis with overall IC/BPS patients of both genders.

The correlation between each urinary biomarker and IC symptoms (ICSI, ICPI, and VAS) and cystoscopic characteristics (MBC and glomerulation) is shown in Table 3. There was a significantly negative correlation between urine levels of CXCL 10, MCP-1, eotaxin, IL-6, MIP-1β, RANTES, TNF-α, and PGE2 and MBC. Moreover, the urine levels of CXCL 10, MCP-1, IL-6, RANTES, and PGE2 were significantly positively correlated with the glomerulation grade. ICSI and ICPI were significantly correlated with urine levels of CXCL 10, BDNF, eotaxin, and IL-6. The VAS score was only correlated with BDNF, IL-6, and PGE2 levels.

When we further subgrouped the non-HIC patients into four subgroups according to the MBC volume (>760 mL or ≤760 mL) or the glomerulation grade (GR > 1 or GR ≤ 1), the urine levels of MCP-1, TNF-α, PGE2 were significantly higher than controls irrelevant with the MBC volume or glomerulation grade. (Table 4) When we subgrouped non-HIC patients according to the combination of MBC and glomerulation grade, all subgroups of patients with non-HIC had significantly higher urinary TNF-α levels than controls irrelevant to the glomerulation grade; whereas patients with an MBC volume of ≤760 mL had significantly higher urinary MCP-1 and PGE2 levels than controls. Patients with a GR of >1 and an MBC volume of ≤760 mL had significantly higher eotaxin and IL-6 levels than controls.

Table 5 shows the ROC results of each urinary biomarker that could be used to predict patients with IC/BPS. Among all biomarkers, TNF-α had the best sensitivity, specificity, PPV, and NPV. Although all biomarkers had a high PPV, all except TNF-α had a low NPV.

Table 6 shows the correlation between urine levels of each cytokine or chemokine and GRA after treatment in patients with IC/BPS. Except BDNF, all urinary biomarkers were significantly correlated with the GRA. A high GRA was correlated with lower urine levels of IL-8, CXCL 10, eotaxin, IL-6, MIP-1β, RANTES, TNF-α, and PGE2. Patients with a high GRA (=2 or 3), which indicated a successful treatment outcome, had significantly lower urinary biomarker levels than those with a low GRA (0 or 1). Figure 2 shows the violin plots of each urinary biomarker between patients with IC/BPS with a high GRA and a low GRA and controls.

## 4. Discussion

The current study showed that IC/BPS can be identified using elevated urine levels of MCP-1, eotaxin, TNF-α, and PGE2. Among the biomarkers, TNF-α showed a higher sensitivity and specificity for diagnosing IC/BPS. Moreover, patients with IC/BPS had significantly higher urine levels of MCP-1, TNF-α, and PGE2 levels than those in controls, irrelevant to the MBC volume or the glomerulation grade. Meanwhile, patients with non-HIC who had a GR of >1 and MBC of ≤760 mL had significantly higher eotaxin and IL-6 levels than controls. Using these urinary biomarkers, we can identify bladder conditions in patients with bladder pain syndrome. Higher urinary cytokine and chemokine levels were significantly associated with a low GRA after treatment. Thus, a higher-grade bladder inflammatory condition may have a less favorable treatment outcome and may require more active treatments. These results indicate that IC/BPS is a syndrome of chronic bladder inflammation resulting from different pathophysiologies, and it increases the inflammation grade and causes different clinical and cystoscopic characteristics.

In general, the etiology of IC/BPS remains unclear, and the condition may be affected by multiple factors, including defective/damaged bladder urothelium, activation of C-fibers, neurogenic inflammation with activation of mast cells, autoimmunity, occult infection, and pudendal nerve entrapment [3,17,18]. Under the inflammatory process in the bladder wall, the detrusor smooth muscle cells could produce cytokines and chemokines that are released into the urine [19]. Therefore, patients with IC/BPS had detectable elevated urinary cytokine and chemokine levels [20]. Gene expression of analysis of urinary sediment could also discriminate patients with IC/BPS from controls [21]. Patients with HIC, but not patients with non-HIC or controls, had upregulated genes, which are mainly associated with inflammation. Urinary protein analysis is a noninvasive approach reflecting the inflammatory status inside the bladder, and it has the potential to developing biomarkers in IC/BPS [12,14,15].

In our cohort, patients with different MBCs and glomerulation grades were equally distributed, indicating that the underlying pathophysiology of each subgroup was heterogeneous. Based on the histopathology classification, chronic inflammation of the bladder wall and urothelial dysfunction varied widely. However, chronic inflammation is one of the most common findings in IC/BPS bladders. The results of this study showed that urine levels of MCP-1, TNF-α, and PGE2 are significantly higher in non-HIC groups than the control group irrelevant to the MBC or the glomerulation grade. MCP-1 has been found to be involved in several inflammatory diseases, including inflammatory bowel disease, allergic asthma, and rheumatoid arthritis, which are closely associated with IC/BPS [22,23]. TNF-α is a proinflammatory cytokine, which could cause inflammation causing bladder damage [24]. The bladder tissue level of TNF-α was significantly increased in patients with HIC. In IC/BPS patients, mast cell activation and the excessive release of TNF-α could elicit an inflammatory response; therefore, the urine level of TNF-α level could increase [25]. PGE2 had been considered a good candidate urinary biomarker for overactive bladder. Nevertheless, the role of PGE2 in the diagnosis of IC/BPS has not been fully clarified. We had previously found a high urinary PGE2 level was significantly associated with a smaller MBC volume and a high glomerulation grade [26]. Taken together, patients with IC/BPS had significantly higher urine levels of inflammatory cytokine and chemokine than controls. Thus, chronic inflammation is the fundamental pathophysiology of IC/BPS.

A high proportion of patients with IC/BPS had multiple somatic pains and functional disorders [27]. A high percentage of patients with IC/BPS had been found to have comorbid autoimmune or neurological diseases [27,28,29]. Moreover, patients with IC/BPS who presented with a large functional bladder capacity are likely to experience depression and irritable bowel syndrome [30]. In this cohort, we found patients with a glomerulation grade of ≤1 and an MBC volume of >760 mL still presented with elevated urinary TNF-α levels, thereby indicating that these patients also had chronic bladder inflammation that caused a systemic inflammatory response to medical or psychological diseases. Therefore, histopathology in this subgroup might have minimal or no evidence of bladder inflammation [9]. Thus, the urine may provide detectable inflammatory cytokines or chemokines, which may explain clinical symptoms in the subgroup of patients with non-HIC.

Currently, IC/BPS is commonly stratified into HIC and non-HIC subtypes according to the clinical characteristics, IC symptoms severity, urodynamic parameters, and different cystoscopic findings [2,3]. Moreover, the treatment strategy and treatment outcome are quite distinct between these two IC subtypes [31]. Based on the study results, significantly higher urine levels of IL-8, CXCL 10, BDNF, eotaxin, and RANTES were noted in HIC than those in non-HIC patients, thereby suggesting the bladder inflammation of HIC is greater than that of non-HIC patients. However, non-HIC patients had significantly higher TNF-α, PGE2, and MCP-1 levels than the controls, regardless of MBC volume, suggesting non-HIC bladders still exhibit a certain degree of chronic bladder inflammation. Although patients with non-HIC had a high MBC and a low glomerulation grade had been considered to have non-bladder-centric IC, characterized by the affect dysregulation and somatic symptoms [31], this study showed that these patients had significantly elevated urinary biomarker levels, which indicated the presence of bladder inflammation. Recently, we found that EB virus-infected T cells in the bladder could be a causative agent of persistent inflammation in patients with IC/BPS. Approximately 46% of IC/BPS bladders presented with evidence of EBV infection. Meanwhile, the proportion of EBV in HIC bladder samples was as high as 88% [32]. There could be different underlying pathophysiologies between HIC and non-HIC bladders that cause different clinical characteristics and changes in urinary biomarker levels.

In a previous study, the cystoscopic characteristics of patients with non-HIC were quite different based on various MBC glomerulation grades [10]. Moreover, patients with non-HIC had significantly elevated urinary MCP-1, TNF-α, and PGE2 levels, irrelevant with the MBC and the glomerulatioon grade. Thus, all non-HIC subgroups somewhat presented with chronic bladder inflammation. In addition, patients with non-HIC who had an MBC of ≤760 mL or a glomerulation grade of >1 had significantly higher urinary eotaxin and IL-6 levels. Eotaxin is considered as a potential urinary biomarker in the diagnosis of patients with IC/BPS [12]. We have previously found that the urinary level of eotaxin had a high specificity for diagnosing ESSIC type 2 IC/BPS [15]. IL-6 is a proinflammatory cytokine and an anti-inflammatory myokine; the urinary level of IL-6 will be higher in patients with HIC, together with the elevated urine levels of IL-8 and CXCL 10 [11]. Therefore, patients with non-HIC who have elevated biomarker levels might also have a higher bladder inflammation grade.

The cystoscopic findings after hydrodistention have been found not significantly correlated with the histological findings showing the degree of bladder inflammation in IC/BPS patients [32]. However, another study reported that a lower MBC volume during hydrodistention under anesthesia could be a marker of a bladder-centric IC/BPS subtype and is associated with higher IC symptom score and significantly greater bladder inflammation [33]. The results of this study found that a small MBC volume was significantly associated with high urinary levels of inflammatory cytokines and chemokines. We also found that only patients with an MBC volume of <760 mL had high urine levels of MCP-1 and PGE2, whereas higher urinary eotaxin and IL-6 levels were observed only in non-HIC patients with an MBC of ≤760 mL and a glomerulation grade of >1. A previous study showed eosinophil and plasma cell infiltration in some patients with IC/BPS bladders, and this finding was in accordance with that of previous human and animal IC/BPS studies [33,34]. All patients with non-HIC have significantly elevated TNF-α levels. However, patients with a higher urinary proinflammatory biomarker level might present with high-grade bladder inflammation with cystoscopic characteristics such as a low MBC volume and a high glomerulation grade. IC/BPS patients that presented with a low MBC volume under anesthesia also had significantly greater inflammation as shown in the higher urine levels of inflammatory cytokines and chemokines. Thus, a low MBC volume is likely associated with a distinct bladder-centric IC/BPS phenotype [9].

In this study, we divided non-HIC into four subgroups, and all non-HIC subgroups had significantly higher urinary levels of inflammatory cytokines and chemokines. Hence, non-HIC bladders are also involved in chronic inflammatory process, and the inflammation causes characteristic IC symptoms including urinary frequency and bladder pain. A recent study showed that some inflammatory mediators in systemic inflammatory diseases might also have important roles in the pathogenesis of IC/BPS [35]. These clinical and proteomic presentations suggest that the non-HIC disease might have chronic inflammation involved both the urinary bladder and other mental factors such as internal conflict or stress disorders [36]. The bladder symptoms of IC/BPS might, in part, result from both the innate bladder inflammation and the effect of systemic medical comorbidities.

Urinary biomarkers for diagnosing IC/BPS and discriminating HIC from non-HIC have been investigated for several decades. However, the results from different studies about the urinary biomarker levels for the differential diagnosis of different IC/BPS subtypes were contrasting, possibly because of the different diagnostic criteria upon patient enrolment, bladder condition upon urine collection, and different disease severities [11]. The study results showed that urinary biomarkers significantly vary among different non-HIC bladder conditions and between non-HIC and HIC bladders. Urinary biomarkers change with time after treatment. Hence, the biomarkers might vary among different bladder conditions in the same patient. Nevertheless, the current study showed that these biomarkers were significantly associated with different bladder conditions and GRA after long-term follow-up. Moreover, this finding shows that urinary biomarkers were not constant and might change with time and were different among IC/BPS with different bladder inflammation. Therefore, these urinary biomarkers are suitable for screening bladder-centric IC/BPS and are useful for the assessment of treatment outcomes.

The limitation of this study is that the patients were enrolled consecutively at our department for further intravesical treatment after the initial diagnosis of interstitial cystitis from 2010 to 2021. In this study, we attempted to investigate the predictive value of urinary biomarkers in the diagnosis and treatment outcome in IC/BPS patients. Therefore, we did not calculate the sample size and power for statistical analysis. That is the reason why the patient number of Hunner’s IC was much lower than the non-Hunner’s IC. In the meantime, the number of control subjects was also small. Another limitation was that the IC/BPS patients were mixed with men and women, but only women were included in the control group. This bias might affect the results of analysis. However, when we analyzed the urine biomarker levels between male and female non-HIC IC/BPS patients, the differences between genders were noted only in three cytokines, and the results did not change the final conclusion.

## 5. Conclusions

Patients with IC/BPS had significantly high urinary MCP-1, eotaxin, TNF-α, and PGE2 levels. The level of other urinary biomarkers such as eotaxin and IL-6 were elevated in patients with non-HIC who had a smaller MBC. High urinary inflammatory cytokine, chemokine, and oxidative stress protein levels reflect a higher bladder inflammation grade and less favorable treatment outcomes.

## Figures and Tables

**Figure 1 biomedicines-10-01149-f001:**
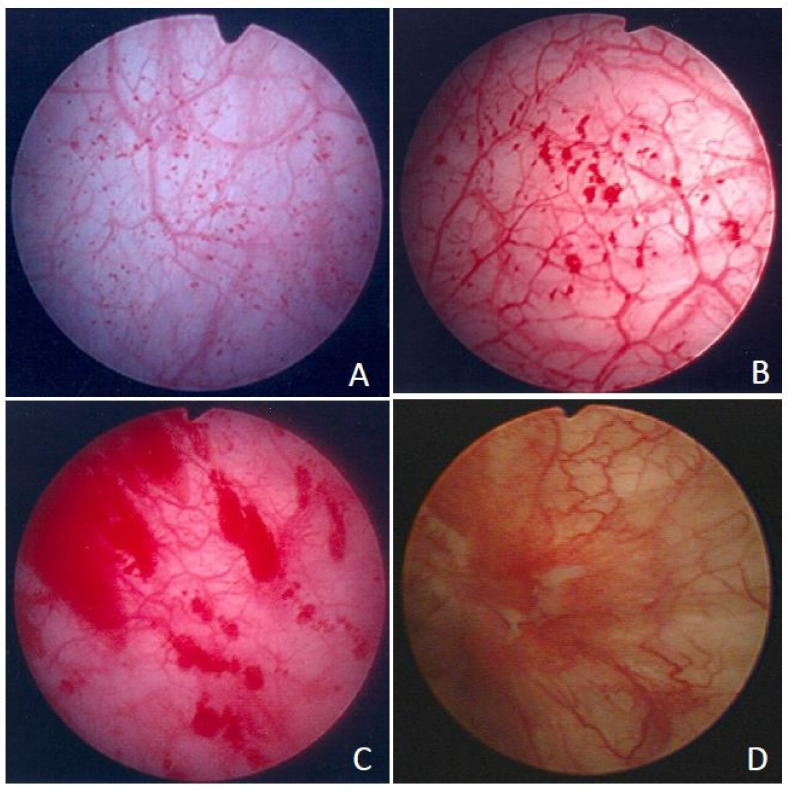
Cystoscopic findings of patients with interstitial cystitis/bladder pain syndrome. (**A**) Grade 0–1 glomerulation with petechiae that developed after hydrodistention, (**B**) grade 2 glomerulation with diffused hemorrhage, (**C**) grade 3 glomerulation with splotch hemorrhage and occasional mucosal fissure, and (**D**) Hunner’s lesion, which can be commonly observed without anesthesia or hydrodistention. The arrows indicate the top of cystoscopic pictures.

**Figure 2 biomedicines-10-01149-f002:**
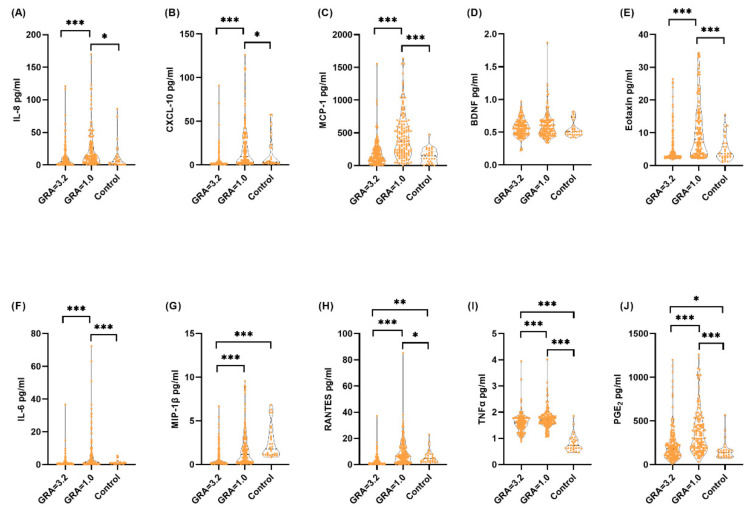
Association between urinary biomarker levels and global response assessment of patients with interstitial cystitis/bladder pain syndrome after treatment. GRA: global response assessment, GRA = 3 significantly improved, GRA = 2: moderately improved, GRA = 1, mildly improved, GRA = 0: not improved. (**A**) Interleukin-8 (IL-8), (**B**) C-X-C motif chemokine ligand 10 (CXCL 10, (**C**) MCP-1, monocyte chemoattractant protein-1 (MCP-1), (**D**) brain-derived neurotrophic factor (BDNF), (**E**) eotaxin-1 (eotaxin-1), (**F**) IL-6, (**G**) macrophage inflammatory protein-1 beta (MIP-1β), (**H**) regulated upon activation, normally T-expressed and secreted protein (RANTES), (**I**) tumor necrosis factor-alpha (TNF-α), (**J**) prostaglandin E2 (PGE2). *: *p* value < 0.05, **: *p* value < 0.01, ***: *p* value < 0.001.

**Table 1 biomedicines-10-01149-t001:** Urinary biomarker levels between overall patients with interstitial cystitis/bladder pain syndrome with and without Hunner’s lesion and controls.

Urine Biomarker	(A) Non-HIC (*n* = 285)	(B) HIC (*n* = 24)	(C) Control (*n* = 30)	*p*-Value	Post Hoc
IL-8	15.9 ± 23.6	34.4 ± 39.7 *	12.5 ± 21.0	0.030	B vs. A,C
CXCL 10	10.1 ± 17.4	35.1 ± 38.2 *	13.8 ± 18.4	0.005	B vs. A,C
MCP-1	299 ± 306 *	289 ± 239 *	147 ± 110	0.001	A,B vs. C
BDNF	0.57 ± 0.14	0.71 ± 0.30 *	0.55 ± 0.12	0.018	B vs. A,C
Eotaxin	7.29 ± 7.05 *	12.0 ± 11.5 *	4.98 ± 3.7	0.017	A,B vs. C
IL-6	2.92 ± 6.96 *	10.8 ± 8.35	1.29 ± 1.35	0.019	A vs. C
MIP-1β	1.18 ± 1.60 *	1.96 ± 2.80	2.52 ± 1.82	0.009	A vs. C
RANTES	5.30 ± 7.90	10.2 ± 10.1 *	6.04 ± 5.15	0.021	B vs. AC
TNF-α	1.65 ± 0.35 *	1.85 ± 0.64 *	0.82 ± 0.33	<0.001	A,B vs. C
PGE2	291 ± 232 *	302 ± 335	161 ± 105	0.037	A vs. C

* *p* < 0.05 compared with the controls; HIC, Hunner’s interstitial cystitis; IL-, interleukin-; CXCL 10, C-X-C motif chemokine ligand 10; MCP-1, monocyte chemoattractant protein-1; BDNF, brain-derived neurotrophic factor; MIP-1β, macrophage inflammatory protein-1 beta; RANTES, regulated upon activation, normally T-expressed, and presumably secreted; TNF-α, tumor necrosis factor-alpha; PGE2, prostaglandin E2.

**Table 2 biomedicines-10-01149-t002:** Urinary biomarker levels between male and female patients with interstitial cystitis/bladder pain syndrome with and without Hunner’s lesion and controls.

Non-HIC Biomarkers	(A) Male (*n* = 45)	(B) Female (*n* = 240)	(C) Total (*n* = 285)	(D) Control (*n* = 30)	*p*-Value A vs. B	*p*-Value A vs. B vs. D	Post Hoc
IL-8	3.87 ± 5.5	18.2 ± 25.0	15.9 ± 23.6	12.5 ± 21.0	<0.001	<0.001	A vs. B
CXCL 10	6.4 ± 7.49	10.8 ± 18.6	10.1 ± 17.4	13.8± 18.4	0.008	0.092	
MCP-1	303 ± 323	298 ± 303	299 ± 36.1	147 ± 110	0.923	0.009	AB vs. D
BDNF	0.58 ± 0.16	0.57 ± 0.13	0.57 ± 0.14	0.55 ± 0.12	0.574	0.638	
Eotaxin	8.53 ± 8.32	7.06 ± 6.78	7.29 ± 7.05	4.98 ± 3.70	0.200	0.070	
IL-6	2.32 ± 4.96	3.03 ± 7.28	2.92 ± 6.96	1.29 ± 1.35	0.526	0.367	
MIP-1β	0.89 ± 0.96	1.23 ± 1.70	1.18 ± 1.60	2.52 ± 1.82	0.189	<0.001	AB vs. D
RANTES	5.18 ± 5.72	5.33 ± 8.26	5.30 ± 7.90	6.04 ± 5.15	0.909	0.880	
TNF-α	1.58 ± 0.23	1.66 ± 0.36	1.65 ± 0.35	0.82 ± 0.33	0.154	<0.001	AB vs. D
PGE2	371 ± 284	276 ± 218	291 ± 232	161 ± 105	0.012	<0.001	AB vs. D
**HIC** **Biomarkers**	**(A) Male** **( *n* = 3)**	**(B) Female** **( *n* = 21)**	**(C) Total** **( *n* = 24)**	**(D) Control** **(*n* = 30)**	***p*-Value** **A v** **s. B**	***p*-Value** **A v** **s. B v** **s. D**	**Post Hoc**
IL-8	16.0 ± 11.0	37.3 ± 42.0	34.4 ± 39.7	12.5 ± 21.0	0.523	0.007	B vs. D
CXCL 10	48.9 ± 59.6	33.4 ± 37.3	35.1 ± 38.2	13.8 ± 18.4	0.573	0.198	
MCP-1	236 ± 244	297 ± 243	289 ± 239	147 ± 110	0.742	0.132	
BDNF	0.57 ± 0.05	0.73 ± 0.32	0.71 ± 0.30	0.55 ± 0.12	0.145	0.003	B vs. D
Eotaxin	8.64 ± 5.02	12.5 ± 12.2	12.0 ± 11.5	4.98 ± 3.70	0.830	0.119	
IL-6	3.83 ± 2.68	11.9 ± 18.5	10.8 ± 17.4	1.29 ± 1.35	1.000	0.056	
MIP-1β	1.08 ± 1.55	2.09 ± 2.95	1.96 ± 2.80	2.52 ± 1.82	0.830	0.058	
RANTES	13.5 ± 17.7	9.78 ± 9.13	10.2 ± 10.1	6.04 ± 5.15	1.000	0.822	
TNF-α	1.63 ± 0.38	1.88 ± 0.67	1.85 ± 0.64	0.82 ± 0.33	0.830	<0.001	AB vs. D
PGE2	465 ± 522	277 ± 311	302 ± 335	162 ± 105	0.268	0.392	

Non-HIC: non-Hunner’s interstitial cystitis; HIC: non-Hunner’s interstitial cystitis; IL-, interleukin-; CXCL 10, C-X-C motif chemokine ligand 10; MCP-1, monocyte chemoattractant protein-1; BDNF, brain-derived neurotrophic factor; MIP-1β, macrophage inflammatory protein-1 beta; RANTES, regulated upon activation, normally T-expressed, and presumably secreted; TNF-α, tumor necrosis factor-alpha; PGE2, prostaglandin E2.

**Table 3 biomedicines-10-01149-t003:** Correlation coefficient between urinary biomarkers and bladder conditions in overall patients with interstitial cystitis/bladder pain syndrome.

	ICSI		ICPI		VAS		MBC		Glomerulation	
	Pearson	*p* =	Pearson	*p* =	Pearson	*p* =	Pearson	*p* =	Pearson	*p* =
IL-8	0.045	0.505	0.004	0.948	−0.042	0.542	−0.092	0.111	0.031	0.586
CXCL 10	0.254	0.000	0.173	0.011	0.092	0.188	−0.238	0.000	0.125	0.032
MCP-1	0.091	0.180	0.037	0.583	−0.063	0.364	−0.253	0.000	0.173	0.003
BDNF	0.189	0.005	0.172	0.011	0.164	0.017	−0.041	0.479	−0.042	0.462
Eotaxin	0.237	0.000	0.144	0.034	0.052	0.456	−0.255	0.000	0.097	0.093
IL-6	0.213	0.002	0.148	0.029	0.207	0.003	−0.231	0.000	0.134	0.020
MIP-1β	0.134	0.048	0.117	0.084	0.003	0.996	−0.187	0.001	0.031	0.598
RANTES	0.207	0.002	0.130	0.054	0.032	0.639	−0.246	0.000	0.132	0.021
TNF-α	0.083	0.220	−0.035	0.600	0.009	0.898	−0.116	0.042	0.082	0.154
PGE2	−0.029	0.668	−0.004	0.955	−0.169	0.015	−0.154	0.007	0.148	0.010

ICSI, Interstitial cystitis symptom index; ICPI, Interstitial cystitis problem index; VAS: visual analog scale; MBC: maximal bladder capacity; IL-, interleukin-; CXCL 10, C-X-C motif chemokine ligand 10; MCP-1, monocyte chemoattractant protein-1; BDNF, brain-derived neurotrophic factor; MIP-1β, macrophage inflammatory protein-1 beta; RANTES, regulated upon activation, normally T-expressed, and presumably secreted; TNF-α, tumor necrosis factor-alpha; PGE2, prostaglandin E2.

**Table 4 biomedicines-10-01149-t004:** Urinary biomarkers in overall patients with interstitial cystitis/bladder pain syndrome with different subtypes and controls.

Urine Biomarker	(A) GR ≤ 1, MBC > 760 (*n* = 85)	(B) GR ≤ 1 MBC ≤ 760 (*n* = 70)	(C) GR > 1 MBC > 760 (*n* = 41)	(D) GR > 1 MBC ≤ 760 (*n* = 89)	(E) Hunner’s IC (*n* = 24)	(F) Control (*n* = 30)	*p*-Value ^#^	*p*-Value ^$^
IL-8	18.7 ± 29.8	16.5 ± 23.6	7.84 ± 10.3	16.5 ± 20.7	34.4 ± 39.7 *	12.5 ± 21.0	0.010	0.011
CXCL 10	6.56 ± 11.6	11.3 ± 20.1	6.08 ± 12.3	14.4 ± 20.5	35.1 ± 38.2 *	13.8 ± 18.4	<0.001	<0.001
MCP-1	204 ± 173	281 ± 276 *	274 ± 294	414 ± 389 *	289 ± 239	147 ± 110	<0.001	<0.001
BDNF	0.57 ± 0.14	0.57 ± 0.14	0.58 ± 0.11	0.55 ± 0.15	0.71 ± 0.30 *	0.55 ± 0.12	0.001	0.001
Eotaxin	6.11 ± 6.42	7.79 ± 7.14	5.48 ± 4.59	8.85 ± 8.12 *	12.0 ± 11.5 *	4.98 ± 3.7	0.002	0.008
IL-6	1.5 ± 2.25	3.47 ± 8.02	2.99 ± 10.1	3.82 ± 7.2 *	10.8 ± 8.35	1.29 ± 1.35	0.008	0.017
MIP-1β	0.9 ± 1.33 *	1.44 ± 1.97	0.8 ± 1 *	1.41 ± 1.69	1.96 ± 2.80	2.52 ± 1.82	0.001	0.058
RANTES	4.06 ± 9.55	5.32 ± 6.54	4.1 ± 5.25	7.05 ± 7.93	10.2 ± 10.1 *	6.04 ± 5.15	0.005	0.005
TNF-α	1.66 ± 0.35 *	1.62 ± 0.27 *	1.6 ± 0.34 *	1.68 ± 0.4 *	1.85 ± 0.64 *	0.82 ± 0.33	<0.001	0.219
PGE2	251 ± 226	284 ± 226 *	265 ± 190	350 ± 252 *	302 ± 335 *	161 ± 105	0.007	0.087

*: *p* <0.05 compared with the controls; ^#^: *p* values between overall IC/ BPS patients and controls; ^$^: *p* values between overall IC/ BPS patients; GR: glomerulation grade; MBC: maximal bladder capacity; IL-, interleukin-; CXCL 10, C-X-C motif chemokine ligand 10; MCP-1, monocyte chemoattractant protein-1; BDNF, brain-derived neurotrophic factor; MIP-1β, macrophage inflammatory protein-1 beta; RANTES, regulated upon activation, normally T-expressed, and presumably secreted; TNF-α, tumor necrosis factor-alpha; PGE2, prostaglandin E2.

**Table 5 biomedicines-10-01149-t005:** Receiver operating characteristic analysis results of each urinary biomarker for predicting interstitial cystitis/bladder pain syndrome.

Urine Cytokines	AUC	Cutoff Value	IC/BPS Sensitivity	IC/BPS Specificity	IC/BPS PPV	IC/BPS NPV
IL-8	0.587	2.100	80.6%	40.0%	93.3%	16.7%
CXCL 10	0.590	1.595	32.7%	90.0%	97.1%	11.5%
MCP-1	0.639	283.1	35.9%	93.3%	98.2%	12.4%
BDNF	0.551	0.543	57.3%	66.7%	94.7%	13.2%
Eotaxin	0.587	12.50	21.0%	96.7%	98.5%	10.6%
IL-6	0.534	0.515	38.2%	83.3%	95.9%	11.6%
MIP-1β	0.774	0.810	60.5%	100%	100%	19.7%
RANTES	0.636	1.495	36.9%	100%	100%	13.3%
TNF-α	0.920	1.050	99.0%	92.6%	98.4%	89.3%
PGE2	0.679	175.4	63.6%	80.0%	97.0%	17.6%

AUC: area under curve; IC/BPS, interstitial cystitis/bladder pain syndrome; IL-, interleukin-; CXCL 10, C-X-C motif chemokine ligand 10; MCP-1, monocyte chemoattractant protein-1; BDNF, brain-derived neurotrophic factor; MIP-1β, macrophage inflammatory protein-1 beta; RANTES, regulated upon activation, normally T-expressed, and presumably secreted; TNF-α, tumor necrosis factor-alpha; PGE2, prostaglandin E2; PPV: positive predictive value; NPV: negative predictive value.

**Table 6 biomedicines-10-01149-t006:** Association between urinary biomarker levels and global response assessment after the treatment of patients with interstitial cystitis/bladder pain syndrome.

	IC/BPS					
Urine Cytokines	(A) GRA = 3 *(n* = 58)	(B) GRA = 2 *(n* = 113)	(C) GRA = 1 *(n* = 109)	(D) GRA = 0 *(n* = 29)	*p*-Value	Post Hoc
IL-8	9.79 ± 18	12.2 ± 17.6	22.1 ± 29.5	32.2 ± 37.2	<0.001	A vs. CD; B vs. C
CXCL 10	2.68 ± 4.06 *	6.02 ± 12.5	17.9 ± 24.4	31.4 ± 28.4	<0.001	AB vs. CD
MCP-1	130 ± 118	199 ± 208	423 ± 330 *	589 ± 380 *	<0.001	A vs. BCD; B vs. CD
BDNF	0.56 ± 0.14	0.57 ± 0.12	0.59 ± 0.19	0.62 ± 0.2	0.301	
Eotaxin	4.09 ± 4.37	5.53 ± 5.17	10.3 ± 8.75 *	13.8 ± 9.12 *	<0.001	AB vs. CD
IL-6	0.78 ± 1.11	1.59 ± 3.93	5.08 ± 9.96 *	10.5 ± 15.4 *	<0.001	AB vs. CD
MIP-1β	0.5 ± 0.88 *	0.91 ± 1.27 *	1.61 ± 2.04	2.69 ± 2.21	<0.001	AB vs. CD
RANTES	1.89 ± 2.62 *	3.17 ± 4.4	7.34 ± 6.52	17.2 ± 16.9 *	<0.001	AB vs. CD; C vs. D
TNF-α	1.51 ± 0.31 *	1.63 ± 0.37 *	1.72 ± 0.39 *	1.9 ± 0.36 *	<0.001	A vs. BCD; BC vs. D
PGE2	171 ± 131	252 ± 222 *	360 ± 248 *	453 ± 309 *	<0.001	A vs. BCD; B vs. CD

* *p* < 0.05 compared with the controls; IC/BPS, interstitial cystitis/bladder pain syndrome; GRA: global response assessment, GRA = 3 markedly improved, GRA = 2: moderately improved, GRA = 1, mildly improved, GRA = 0: not improved; IL-, interleukin-; CXCL 10, C-X-C motif chemokine ligand 10; MCP-1, monocyte chemoattractant protein-1; BDNF, brain-derived neurotrophic factor; MIP-1β, macrophage inflammatory protein-1 beta; RANTES, regulated upon activation, normally T-expressed, and presumably secreted; TNF-α, tumor necrosis factor-alpha; PGE2, prostaglandin E2.

## Data Availability

Data are available if contact the corresponding authors.

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
