# Peer review of "Use of Urinary Cytokine and Chemokine Levels for Identifying Bladder Conditions and Predicting Treatment Outcomes in Patients with Interstitial Cystitis/Bladder Pain Syndrome"

_biomedicines, 2022, doi:10.3390/biomedicines10051149_

Round 1

Reviewer 1 Report

I suggest to improve the methods section; I suggest to add sample size calculation and statistical power, I think it might improve the section and the manuscript.

Author Response

Comments and Suggestions for Authors: I suggest to improve the methods section; I suggest to add sample size calculation and statistical power, I think it might improve the section and the manuscript.

Reply: Thank you for the comment. The patients were enrolled consecutively at our department for further intravesical treatment after the initial diagnosis of interstitial cystitis from 2010 to 2021. In this study, we attempted to investigate the predictive value of urinary biomarkers in the diagnosis and treatment outcome in IC/BPS patients. Therefore, we did not calculate the sample size and power for statistical analysis. That is the reason why the patient number of Hunner’s IC was much lower than the non-Hunner’s IC. In the meantime, the number of control subjects is also small, that could also be the reason why the negative predictive values of most cytokines are not high. We have added a statement in the limitation of the study. (Lines 408 to 415)

Reviewer 2 Report

This study evaluated the role of urinary cytokine and chemokine levels for identifying bladder conditions and predicting treatment outcomes in patients with interstitial cystitis/Bladder pain syndrome. They found that patients with IC/BPS had significantly high urinary MCP-1, eotaxin, TNF-α, and PGE2 levels and most urinary biomarkers were significantly associated with MBC, glomerulation grade, and treatment outcome. Length and readability are good. The choice of topic is interesting because IC/BPS is a chronic condition with serious consequences in patient’s mental and psychological health. The study conducted well and the appropriate statistical tests were performed. The results are presented in a logical manner and the discussion flows well. The manuscript is well organized and written. The only drawback of the study (if we can say so), is that the causes of IC/BPS are more than inflammatories and recently many papers correlates this chronic condition with pudendal nerve entrapment and with other neurological conditions. 

In my opinion this paper would be possible accepted without any revisions

Author Response

Reply: Thank you for the comment. The pathophysiology of IC/BPS is multifactorial. Based on the histopathological studies, neurogenic inflammation could also play an important role, which might originate from innate bladder inflammation or pudendal nerve entrapment. We have discovered a group of women who had definite pelvic floor muscle tenderness possibly due to pudendal nerve entrapment and could have pelvic pain relief after pelvic floor muscle massage. Interestingly, the urine cytokines levels are not different from those without pelvic floor pain syndrome. A statement of pudendal nerve entrapment and a reference have been added in the revised manuscript. (Lines 278 and reference 18)

Reviewer 3 Report

The current manuscript describes and will probably establish the use of Urinary Cytokine and Chemokine Levels for the Identification of Bladder Conditions and Prediction of Treatment Outcomes in Patients with Interstitial Cystitis/Bladder Pain Syndrome. The authors did a good research with a novel hypothesis. I think the manuscript will be more improved after the following changes.

  1. Abstract is too long, please rewrite and keep it within the range.
  2. Some references were cited after a full stop (.), for example, line 94, 104. Please cite properly.
  3. Figure 1, all images were not full image, upper and lower portions are cut down. Please try to provide all similar size images. What is the arrow cut in the Figure1B-D? Please clearly mention the differences in the images near them.
  4. Total 309 patients, 261 women and 48 men. 30 controls all were women. Please mention why there was no man control. And also please show the data separately (men and women) in the tables along with women controls for better understanding and to know if there are any differences. If possible, show the table data in graphs for better visualization and easier illustration.
  5. Please include a graphical abstract or a summary figure for a better illustration of the study results.

Author Response

1. Abstract is too long, please rewrite and keep it within the range.

Reply: Thank you for the comment. We have reduced the length of abstract. (Lines 29 to 53)

2. Some references were cited after a full stop (.), for example, line 94, 104. Please cite properly.

Reply: Thank you for the comment. We have revised the citation of the references properly.

3. Figure 1, all images were not full image, upper and lower portions are cut down. Please try to provide all similar size images. What is the arrow cut in the Figure1B-D? Please clearly mention the differences in the images near them.

Reply: Thank you for the comment. The figure 1 has been revised to demonstrate the full image of cystoscopic findings of Hunner’s IC and non-Hunner’s IC. (Line 580)

The arrow in the top of image is a pointer for the top of cystoscopic picture. (Line 578-579)

4. Total 309 patients, 261 women and 48 men. 30 controls all were women. Please mention why there was no man control. And also please show the data separately (men and women) in the tables along with women controls for better understanding and to know if there are any differences. If possible, show the table data in graphs for better visualization and easier illustration.

Reply: Thank you for the comment. In this study we did not collect urine from normal male controls because it was not allowed by IRB to perform videourodynamic study in men without lower urinary tract symptoms. Using female pure stress urinary incontinence as controls is a convenient way to obtain control data of normal bladder without bladder outlet obstruction and urine samples for comparison. We have analyzed the female and male data of each subgroup of IC/BPS as in the new table 2 (page 21). When we divided patients into male and female IC/BPS patients, and compared the urine biomarkers between male and female patients, a lower level of IL-8 and CXCL 10 and a higher level of PGE2 were found in male non-HIC patients than those in female patients. However, there was no significant difference in all urine biomarkers between male and female HIC patients. (Table 2) The differences of urinary levels of biomarkers compared with those of controls were not different between overall and male or female IC/BPS patients. Therefore, we did further analysis with overall IC/BPS patients of both genders. (Lines 216 to 223)  A statement of study limitation has been added in the text. (Lines 415-419)

5. Please include a graphical abstract or a summary figure for a better illustration of the study results.

Reply: Thank you for the comment. The graphic abstract will be provided after the paper has been accepted for publication.